# A Time-Course Study of the Expression Level of Synaptic Plasticity-Associated Genes in Un-Lesioned Spinal Cord and Brain Areas in a Rat Model of Spinal Cord Injury: A Bioinformatic Approach

**DOI:** 10.3390/ijms22168606

**Published:** 2021-08-10

**Authors:** Vito Antonio Baldassarro, Marco Sanna, Andrea Bighinati, Michele Sannia, Marco Gusciglio, Luciana Giardino, Luca Lorenzini, Laura Calzà

**Affiliations:** 1Department of Veterinary Medical Science, University of Bologna, Via Tolara di Sopra, 50, 40064 Ozzano Emilia, Italy; vito.baldassarro2@unibo.it (V.A.B.); andrea.bighinati@unibo.it (A.B.); luciana.giardino@unibo.it (L.G.); luca.lorenzini8@unibo.it (L.L.); 2Interdepartmental Center for Industrial Research in Life Sciences and Technologies, University of Bologna, Via Tolara di Sopra, 41/E, 40064 Ozzano Emilia, Italy; marco.sanna9@unibo.it (M.S.); michele.sannia2@unibo.it (M.S.); 3IRET Foundation, Via Tolara di Sopra, 41/E, 40064 Ozzano Emilia, Italy; marco.gusciglio@outlook.it; 4Department of Pharmacy and BioTechnology, University of Bologna, Via San Donato, 15, 40127 Bologna, Italy; 5Montecatone Rehabilitation Institute, 40026 Imola, Italy

**Keywords:** spinal cord injury, rat, transcriptomic, bioinformatic analysis

## Abstract

“Neuroplasticity” is often evoked to explain adaptation and compensation after acute lesions of the Central Nervous System (CNS). In this study, we investigated the modification of 80 genes involved in synaptic plasticity at different times (24 h, 8 and 45 days) from the traumatic spinal cord injury (SCI), adopting a bioinformatic analysis. mRNA expression levels were analyzed in the motor cortex, basal ganglia, cerebellum and in the spinal segments rostral and caudal to the lesion. The main results are: (i) a different gene expression regulation is observed in the Spinal Cord (SC) segments rostral and caudal to the lesion; (ii) long lasting changes in the SC includes the extracellular matrix (ECM) enzymes *Timp1*, transcription regulators (*Egr*, *Nr4a1*), second messenger associated proteins (*Gna1*, *Ywhaq*); (iii) long-lasting changes in the Motor Cortex includes transcription regulators (*Cebpd*), neurotransmitters/neuromodulators and receptors (*Cnr1*, *Gria1*, *Nos1*), growth factors and related receptors (*Igf1*, *Ntf3*, *Ntrk2*), second messenger associated proteins (*Mapk1*); long lasting changes in Basal Ganglia and Cerebellum include ECM protein (*Reln*), growth factors (*Ngf*, *Bdnf*), transcription regulators (*Egr*, *Cebpd)*, neurotransmitter receptors (*Grin2c*). These data suggest the molecular mapping as a useful tool to investigate the brain and SC reorganization after SCI.

## 1. Introduction

An acute lesion of the Spinal Cord (SC) is followed by an ensemble of reactive phenomena, embracing the molecular, structural, electrophysiological and functional levels, which involve not only the lesion site, but all the Central Nervous System (CNS) areas anatomically related and even unrelated to the lesion. The molecular, anatomical and functional changes extend over a long time-span, from the trauma to the consolidated final outcome, and might be responsible for partial or complete functional recovery, but also for aberrant phenomena, such as motor patterns supporting spasticity and sensory patterns supporting chronic neuropathic pain [1]. For example, electrophysiological studies have suggested that neuronal circuits below the SC level of injury, thus deprived of supra-spinal inputs, undergo progressive and long-lasting remodeling [2], being this a possible source of neuronal dysfunctions [1,3].

On the other hand, while anatomical and functional restoration at the lesion site is strongly limited by the poor repair capability of the CNS and by the anatomical disruption that follows traumatic lesions, the overall recovery of a complex function, such as locomotion, involves the recruitment of alternative or dormant pathways and learning processes [4]. In fact, an alteration of the resting-state of sensorimotor network and an increased connectivity between motor components has been demonstrated in patients with Spinal Cord Injury (SCI) in several task-evoked functional magnetic resonance imaging (fMRI) studies [5], and neuroplasticity is currently a focus for rehabilitation approaches [6].

“Neuroplasticity” includes multiple levels of organization, from the molecular level in spinal and supraspinal circuits, to behavioral and functional performance [7]. Even though neuroplasticity is regarded as the biological substrate of rehabilitative motor training, and eliciting it represent one of the most widely used approaches to promote moderate recovery following injuries of the CNS, few preclinical data useful for translational purposes are available to support and provide experimental evidence on undergoing molecular and structural events [8].

One of the early events following SCI is a change in gene transcription in the CNS, that supports functional adaptation and compensation to the lesion, from early-activated to genes supporting long-lasting changes in neuronal function. The study of this complexity, that includes dozens of genes, their temporal expression profile in respect to the lesion and several CNS areas also far from the lesion, can be supported by “omics” technologies and bioinformatic data analysis and integration. However, most of the published data are based on available databases [9,10,11,12] or focused on SC tissue sampled at the lesion site at early time points [13,14,15], thus not considering the complex regional and temporal pattern of neural plasticity.

In attempt to fill this gap and highlight the molecular substrate of neuroplasticity in SCI, in this study we investigated changes in the expression level of genes associated with synaptic plasticity at 24 h, 8 and 45 days after contusive SCI in rats Motor Cortex (CTX-M), Basal Ganglia (BG), Cerebellum (CB) and un-injured SC, rostral and caudal to the lesion level. We adopted a data-driven approach starting from own experimental data obtained by animals fully characterized in term of clinical state, locomotor behavior and anatomy of the lesion, and a bioinformatic analysis set on stringent statistical criteria. The investigated genes include Immediate-Early (IEGs) and Late Response Genes, Long Term Potentiation (LTP) and Long Term Depression (LTD) associated genes, genes encoding for Cell Adhesion Molecules, Extracellular Matrix (ECM) Molecules, CREB Cofactors, Neuronal Receptors and Postsynaptic Density Proteins (PSD).

## 2. Results

### 2.1. Lesion Characterization

The lesion was characterized for the functional outcome and anatomical landmarks. Animals were monitored for body weight (Figure 1A), Basso, Beattie and Bresnahan (BBB) score (according to [16], Figure 1B) and recovery of the spontaneous bladder function, expressed as percentage of animals showing impairment of spontaneous urination, in which 100% correspond to the total number of animals (Figure 1C). As expected, lesioned rats have a lower body weight gain (two ways ANOVA, *p* < 0.0001). Bladder function recovers between 7 and 10 days post lesion (DPL) (Figure 1C). Locomotion, as evaluated by the BBB score having 21 as maximum score in all animals before surgery, partially recovers, thus stabilizing at 14 days.

Gait analysis was also performed by a computerized video-tracking apparatus, and the most significant parameters are presented in Figure 1D,E. Overall, lesioned rats show severe alteration in the gait pattern. For example, the step sequence regularity index, that is calculated as % index for the degree of interlimb coordination during gait (number of normal step sequence patterns (NSSP), multiplied by the number of paws and divided by the number of paw placements; RI = 100% × (NSSP × 4) / no. of paw placements) is severely altered (Figure 1D), as also the duty cycle for both, hind and front paws (Figure 1E,F). The duty cycle (%) is the ratio of stand time to step cycle (duty cycle = stand time/step cycle). Stand time (s) is calculated by the duration of contact of each paw with the walkway. Step cycle (s) is calculated by the duration of two consecutive initial contacts of each paw (step cycle = stand time + swing time). These parameters show the different temporal dimensions (duty cycle), and interlimb coordination (RI) in sham and lesioned animals.

We then characterized the anatomy of the lesion at 8 and 45 DPL, and results are presented in Figure 2. Panel A indicates the site of the lesion. The SCs were serially sectioned according to the coronal plane from level T6 to level L3; alternate sections were stained with toluidine blue and HE, and the area of lesion measured at defined space intervals along the contused area. The resulting 3D reconstruction at 45 DPL is presented in Figure 2B, were the histograms express the lesioned area as % value of the total area. The schema in panel C reports the anatomical distribution of ascending (left side) and descending (right side) pathways, and the low-power micrographs in D illustrate the coronal section along the lesion extension. To better appreciate the dorso-ventral extension of the lesion, SC was also horizontally sectioned at 8 and 45 DPL. The resulting 3D reconstruction is presented in Figure 2E,F, respectively. The overall analysis indicates that the lesion determines a severe disruption of the dorsal part of the SC over a rostro-caudal extension 3 mm (Figure 2B, histogram) and a larger extension of the lesion caudally to the impact epicentrum. The ventral structures, although invested by the contusion, are less extensively destroyed and the histological analysis reveals, as expected, an intense infiltration of inflammatory cells, areas of hemorrhages and necrosis at 8 DPL (Figure 2G,H). Phagocytic cells (gitter cells), having spherical shape with bubbly margin, reduced nucleus, lipid-laden, are also evident (arrows in H). At 45 DPL, cavitation is observed in communication with the central canal (Figure 2I) and surrounded by an ependymal-like layer (Figure 2J). Inflammatory and gitter cells are also present (Figure 2K).

### 2.2. mRNA Expression Level of Synaptic Plasticity Associated Genes

The mRNA expression analysis has been carried out using the Qiagen RT^2^-qPCR array for the expression of genes involved in synaptic plasticity according the MIQE Guidelines [17], having reference genes automatically detected by the analysis software. The list of investigated genes is reported in Appendix A. For biological averaging and variance reduction, samples from each group were pooled for microarray experiments, since pooling dramatically improves accuracy for very small designs [18,19,20]. Five animals for each time-point and 5 control, unlesioned rats, were included in the study. Regulated genes were selected by relative Fold Change (FC) ≥ 2. To analyze the temporal evolution of gene expression, data from lesioned animals were normalized *vs* intact at all the time points (1, 8 and 45 DPL).

We first analyzed the SC rostral and caudal to the lesion, and a distal cervical segment. Results are presented in Figure 3, were each graph reports the FC at different time-points (DPL), with their numeric values in tables. In the rostral segment, seven genes (*Cebpd*, *Fos*, *Junb*, *Mmp9*, *Ngf*, *Timp1*, *Tnf*) resulted upregulated at 8 DPL, and only *Timp1* remains upregulated at 45 DPL. Seven genes (*Camkg2*, *Egr3*, *Gnai1*, *Grm3*, *Grm8*, *Igf1*, *Nr4a1*) were downregulated at 1 DPL, and *Nr4a1* and *Egr3* remain downregulated at 45 DPL. In the portion of the SC caudal to the lesion, six genes resulted upregulated (*Cebpb*, *Cebpd*, *Junb*, *Mmp9*, *Timp1*, *Tnf*) with only *Timp1* overexpressed at all considered times points. The downregulated genes resulted differentially distributed across the considered times, with *Egr3*, *Grm8* and *Igf1* emerged at 1 DPL; *Arc*, *Egr4*, *Ntf3* at 8 DPL; *Gnai1* at 45 DPL and only *Nr4a1* downregulated at all the considered DPLs.

When considering the cervical SC, that is relatively far from the lesion site, three genes resulted upregulated at 1 DPL (*Cebpd*, *Cnr1* and *Timp1*), being *Timp1* still overexpressed at 8, but not at 45 DPL. Few genes resulted downregulated early from the lesion (1 and 8 DPL), including *Bdnf*, *Egr3*, *Egr4*, *Ngfr*, *Nr4a1* and five genes (*Egr1*, *Egr2*, *Egr3*, *Egr4*, *Nr4a1*) are downregulated at long time (45 DPL).

We then analyzed gene expression in the CTX-M (Figure 4). This area shows few upregulated genes at early times *(Cebpd*, *Cnr1*, *Gria1*, *Kif17)*, which remain upregulated at late time (*Cebpd*, *Cnr1*, *Gria1)*, and several downregulated genes (*n* = 17). *Igf1*, *Ntrk2* (*Trkb*), *Prkg1* and *Ywhaq* are downregulated at all time points, while *Mapk1*, *Nos1* and *Ntf3* remain downregulated at 45 DPL. A downregulation restricted to the early response to the lesion (1 DPL) occurs for *Grm2*, *Grm3*, *Ngfr* (*p75^NTR^*), *Klf10*, *Mmp9*, *Pcdh8* and *Prkcg*; while three genes are downregulated only 8 DPL (*Cebpb*, *Egr2* and *Fos*).

We finally analysed the full gene set for synaptic plasticity in the BG and CB (Figure 5). BG showed three upregulated (*Ephb2*, *Grm2*, *Reln*) and two downregulated (*Bdnf*, *Egfr2*) genes (Figure 5A); while the CB analysis revealed that *Ngf* gene is the only one upregulated at 45 DPL and four genes were downregulated (*Cebpb*, *Egr3*, *Fos*, *Grin2c*) (Figure 5B).

### 2.3. Pathway Enrichment Analysis

We then adopted a bioinformatic approach to identify gene clustering in the different areas and at the different time points. Time-series Gene Set Enrichment Analysis were performed by comparing the Pearson correlation between post-injury genes expression profiles, from the different CNS areas and three ideal expression profiles: an early profile peaking at 1 DPL, an early profile peaking at 8 DPL and a late profile peaking at 45 DPL. These time points correspond to acute, sub-acute and chronic injury phases. The complete list of pathways resulting from GSEA analysis are available in Appendix A).

Significant results from the SC segments (False Discovery Rate (FDR) ≤0.1) are summarized in Figure 6, and relative data are presented in Table 1 and Table 2 (corresponding pathways Enrichment plots and gene set Heatmaps are reported in Appendix A).

The SC segment rostral to the lesion showed an early (1 DPL peak) upregulation of inflammation and differentiation pathways: interleukin-4 and interleukin-13 signaling (R-RNO-6785807), signaling by interleukin (R-RNO-449147), TNFα signaling (M5890), negative regulation of peptidase activity (GO:0010466), negative regulation of cysteine type endopeptidase activity (R-RNO-2000117), osteoclast differentiation (GO:0030316), myeloid cell differentiation (GO:0002573), myeloid leukocyte differentiation (GO:0030099), transcriptional regulation of white adipocyte differentiation (R-RNO-381340) and ossification (GO:0001503); together with a downregulation of glutamate receptor signaling pathway (GO:0007215). Interlukin-4 and interleukin-13 signaling pathway (R-RNO-6785807) were the only pathways upregulated at 8 DPL.

The SC segment caudal to the lesion is characterized by an early enrichment for inflammatory pathways (R-RNO-6785807, R-RNO-449147, GO:0019221, GO:0010466) and a downregulation of neuronal system (R-RNO-112314, R-RNO-112315, R-RNO-112316), cell-cell signaling (GO:0099536), AMPA receptors trafficking (R-RNO-399719), activation of NMDA receptors (R-RNO-442755) and G-alpha I signaling events (R-RNO-418594) pathways. At 8 DPL, interleukin-4 and interleukin-13 signaling (R-RNO-6785807) and negative regulation of peptidase activity (GO:0010466) pathways remained upregulated; while a downregulation of signaling by GPCR (R-RNO-372790), regulation of synaptic plasticity (GO:004816), regulation of trans-synaptic signaling (GO:0099177), behavior (GO:0007610), sensory perception (GO:0007600), response to ethanol (GO:0045471) and most of the 1 DPL pathways (R-RNO-112314, R-RNO-112315, R-RNO-112316, GO:0099536, R-RNO-442755) remained downregulated. When looking for late regulations (45 DPL peak), only the SC rostral segment showed significant results, with an upregulation of cell-cell signaling pathway (GO:0099536) and regulation of trans synaptic signaling pathway (GO:0007267); a downregulation of TNFα signaling (M5890), mononuclear cell differentiation (GO:1903131), immune system development (GO:0002520), apoptosis (M5902), NGF stimulated transcription (R-RNO-9031628), positive regulation of biosynthetic process (GO:0009891), positive regulation of nucleobase containing compound metabolic process (GO:0045935) pathways and various others related to positive RNA polymerase II transcription, mostly pri-miRNA transcription (GO:0061614, GO:1902895, GO:0045944), was also detected.

While the analysis of SC cervical segment did not produce significant result, the analysis of cerebral CTX-M, BG and CB samples, showed distinct profiles, summarized in Figure 7 and represented in Table 3, Table 4 and Table 5.

The CTX-M is characterized by an early 1 DPL downregulation of regulation of response to external stimulus (GO:0032101), regulation of peptide transport (GO:0090087), positive regulation of organelle organization (GO:0010638), activation of MAPK activity (GO:0000187), positive regulation of MAPK cascade (GO:0043410) and MAP kinase activity (GO:43406), positive regulation of cell population proliferation (GO:0008284), muscle cell proliferation (GO:0033002), positive regulation of locomotion (GO:0040017), regulation of cell-cell adhesion (GO:0022407) and ossification (GO:0001503) pathways. At 8 DPL a downregulation of muscle cell proliferation (GO:0033002) is still present, together with the downregulation of positive regulation of biosynthetic process (GO:0009891), positive regulation of cell population proliferation (GO:0008284) and NGF stimulated transcription (R-RNO-903162) pathways.

The BG and CB are characterized by late (45 DPL) regulations; the first showing an upregulation in cell-cycle and cell growth (GO:0040007, GO:0007049), negative regulation of gene expression (GO:0010629), post-transcriptional regulation of gene expression (GO:0010608), cellular metabolic amide and peptide related (GO:0043603, GO:0006518) and gamete generation (GO:0007276) pathways; the latter a downregulation of positive regulation of synaptic transmission (GO:0050806), and NMDA receptor regulation (R-RNO-442755, R-RNO-438066, R-HSA-9617324) pathways. A downregulation of response to temperature stimulus was detected at 8 DPL in the CB (GO:000926). GSEA leading edge analysis on these sets of data, showed concordant results with the DE genes analysis and provided hints on additional deregulated genes that were not selected by FC ≥ 2 filter (Appendix A).

## 3. Discussion

The study of the spatial and temporal pattern of gene expression regulation in experimental animals is a key step in understanding the molecular bases of neuroplasticity associated to SCI, helpful to understand maladaptive plasticity supporting neuropathic pain and spasticity, and to map spontaneous recovery and neurorehabilitation interventions in a translational perspective [21,22].

In this proof-of-concept study we investigated the expression regulation of genes involved in synaptic plasticity also exploiting a data-driven approach for the analysis. We included in the study brain and SC areas outside the lesioned segment, i.e., in the CTX-M, BG, CB and SC segments rostral (T4−T7) and caudal (T12−L3) to the lesion (T9). Tissue samples were collected at 3 different time points, corresponding to acute injury phase (1 DPL), in which vascular damage, ionic imbalance, neurotransmitter accumulation (excitotoxicity), free radical formation, calcium influx, lipid peroxidation, inflammation, edema and necrotic cell death take place [23,24]; (ii) sub-acute phase (8 DPL), involving apoptosis, demyelination of surviving axons, Wallerian degeneration, axonal dieback, matrix remodeling and evolution of a glial scar around the injury site [25]; (iii) chronic phase (45 DPL), with the formation of a cystic cavity, progressive axonal die-back, maturation of the glial scar and stabilization of the functional recovery [26]. The inclusion of SC segments which are not directly involved in the primary lesion, brain areas responsible for the cortical, extrapyramidal and cerebellar movement control, as well as the inclusion of a late time point, together with the focus on synaptic plasticity genes, represents the novelty of this study, covering a still void segment in literature.

We first carefully characterized the animal model from both a functional and an anatomical point of view. Since SCI induces a long-lasting disruption of the estrous cycle [27], we used female rats in which the estrous stages were not determined. The full characterization of the lesion model is, in fact, a prerequisite for result robustness and reproducibility in all experiments on SCI [28]. In particular, it is important to determine if the lesion model induces a full anatomical disconnection between the rostral and the caudal segments of SC at the site of injury. In the contusion model used in this study, the lesion induced a severe disconnection, which preserves only the external part of the white matter in the ventral horn, where ascending spinothalamic and spinocerebellar, and part of the descending corticospinal and vestibulospinal descending tract pass [29]. This could explain the partial, spontaneous locomotor recovery observed in our rats.

We then mapped the expression regulation of synaptic plasticity-associated genes. Since gene expression regulation is a key step in neural plasticity, especially regarding long-lasting changes, we used a pathway array approach including almost 90 genes. The most relevant results, as summarized in Figure 8, are the following: (i) a different gene expression regulation is observed in the SC segments rostral and caudal to the lesion; (ii) CTX-M is the brain area showing more gene expression changes; (iii) long lasting changes in the SC include the ECM enzymes *Timp1* [30], transcription regulators (*Egr*, *Nr4a1*), second messenger associated proteins (*Gna1*, *Ywhaq*); (iv) long-lasting changes in CTX-M includes transcription regulators (*Cebpd*), neurotransmitters/neuromodulators and receptors (*Cnr1*, *Gria1*, *Nos1*), growth factors and related receptors (*Igf1*, *Ntf3*, *Ntrk2*), second messenger associated proteins (*Mapk1*); long lasting changes in BG and CB include ECM protein (*Reln*), growth factors (*Ngf*, *Bdnf*), transcription regulators (*Egr*, *Cebpd)* and neurotransmitter receptors (*Grin2c*).

The pathway enrichment analysis of data derived from the SC suggests that the early profile (1 and 8 DPL) involves differentiation-, interleukin mediated-signaling paths and glutamate receptor-related path, while the late profile (45 DPL) is characterized by regulation of RNA transcription and synaptic cell signaling, with a silencing of inflammation-related pathways.

The late time point included in this study reflect the ongoing molecular events supporting the synaptic function. In fact, while input deprivation induces an early decline of activity in the brain areas, it is followed by a reorganization of the receptive fields, probably based on a re-balance between inhibitory and excitatory synaptic changes that requires the lesion stabilization [31]. Being the expression of genes related to synaptic plasticity “activity-dependent”, we assume that the abrupt interruption of axonal connections [32] can trigger rapid and long-lasting gene expression changes in the neurons originating axons of the descending/ascending spinal tracts, but also trans-synaptically [33].

At 45 DPL, same genes are differentially expressed rostrally and caudally to SCI site. In particular, only *Timp1* remains up-regulated, as previously described [30], and *Nr4a1* down-regulated in both segments; while *Egr3* and *Gnai1* are downregulated in the rostral and caudal segment, respectively. The different gene members of the EGR family of transcriptional regulators are involved in CNS function, and *Egr3* is required specifically for short-term memory [34]. *Gnai1* belong to the family of heterotrimeric signal-transducing molecules consisting of alpha, beta and gamma subunits, and the encoded proteins are part of a complex that responds to beta-adrenergic signals. This gene also emerged as hub gene in a protein-protein interaction network based on Gene Expression Omnibus (GEO) database, including 1, 3, 7 and 14 DPL [35]. While several preclinical and clinical studies indicated substantial differences in the anatomical outcome of the SC segments rostral and caudal to the lesion [36,37], the supporting mechanisms are not clear. The secondary injury extension is not specular respect to the lesion epicentrum. In a compression model, a different energy metabolism related to vasculature dynamics has been described, suggesting that a deficit in the glycolytic pathway accelerates the caudal degeneration; while immediate rostral degeneration is exacerbated by oxidative stress [38]. The extension of the damage outside the secondary lesion area, shows a different anatomical evolution according to the rostro-caudal gradient, including a different white vs. gray matter degeneration related to the distance of the axonal lesion from the cell body. Moreover, different sensory, motor and intraspinal pathways are differentially vulnerable [39,40]. Due to the strong correlation between anatomical and functional outcomes, a more detailed study of the rostro-caudal impact of the SCI is desirable.

CNS brain areas involved in the motor function (CTX-M, BG and CB) are characterized by a profound gene expression regulation at both early and late phases. In particular, two genes were up-regulated, and 24 down-regulated, in CTX-M at the early phase; thus confirming at molecular level the electrophysiological and transcranial optogenetic mapping data, describing an initial loss of motor map (early phase) and a subsequential partial recovery (late phase) [31,41,42]. The regulated gene expression include inflammatory genes and a downregulation of *Mapk2*, as described at 14 DPL in a whole brain analysis [43]. In the late phase, 3 genes are up-regulated and 6 down-regulated, and the pathway analysis indicates the NMDA transmission regulation as core of the regulated net. NMDA is a glutamate receptor and ion channel found in neurons, possibly mediating the increased excitability of the CTX-M. The elective activation of glutamate neurons in the primary CTX-M was related to the functional prognosis of patients and can promote functional recovery after SCI [44]. Other regulated genes not directly related to the NMDA net includes *Cnr1* (Cannabinoid Receptor 1), which is involved in sensory deprivation-induced cortical plasticity by mediating long-term potentiation and depression of synapse strength and the fine-tune of excitatory/inhibitory balance [45]. Several growth factor- and growth factor receptor encoding genes are down-regulated, e.g., *Igf1*, *Ntf3*, *Ntrk2*. IGF-1 affects the size of cortical receptive fields and the cutaneous threshold and is implied in cortical changes due to hypoactivity [46]; *Ntf3* is among the differentially expressed genes in the auditory cortex following auditory deafferentation [47]. *Ntrk2* encodes for Brain Derived Neurotrophic Factor (BDNF) high affinity receptor, TrkB, being BDNF a key growth factor for cortical neuroplasticity following lesions and physical exercise [48]. These results are not surprising, considering that learning processes involving BDNF in several spinal and supra-spinal circuits, induced by training and experience, can promote recovery after SCI [4]. In addition, human studies have described motor cortical maps reorganization after SCI [49,50,51]. These studies have also provided the background and the appropriate endpoint to evaluate efficacy of neuro-rehabilitation and neurostimulation protocols [52], evoking “neuroplasticity” as possible mechanism [7]. The gene expression regulation that we observed in our experiment possibly reflects the deep reorganization of motor pathways that starts immediately after injury [53,54].

Notably, BDNF encoding gene is upregulated at 45 DPL in the BG, while NGF encoding gene is upregulated in the CB. BDNF is a major player in the BG biology and function, also mediating various synaptic reorganization processes [55]. Moreover, BG and all extrapyramidal pathways undergo extensive reorganization changes, the magnitude of which predicts the functional recovery after SCI [56]. NGF endogenous levels increases in the CB in experimental conditions stimulating neuroplasticity (environmental enrichment) also after lesions [57], by modulating glutamatergic transmission [58].

In the context of transcriptomic mapping for the SCI, bulk RNA-seq and single cell RNA-seq (scRNA-seq) are probably the most powerful tools to build a complete picture of the complex cellular microenvironment dynamics. When bulk RNA-seq is used in a rat model of SCI analyzed at early (1, 6 DPL) and late (28 DPL) phases, the pathway enrichment analysis revealed a prevalence of immunoresponse, but the respective contribution of the resident and infiltrating immune system cells to the lesion progression can’t be established [59]. More information can be obtained by scRNA-seq, but most of these studies have been performed on mouse models, which show huge differences with rats, especially in cavitation and scar formation, and is less similar to human SCI progression [60]. ScRNA-seq in mouse SCI acute (1, 3 DPL) and sub-acute (7 DPL) indicated that the main response is mediated by resident and infiltrating immune system and OPC activation [61]. This result is in line with our observations of the activation of ECM reorganization genes in the context of the synaptic plasticity pathway. Interestingly one of the main genes involved in recovery, *Igf1*, identified in mouse SCI scRNA-seq, resulted downregulated in the acute phase in our rat model [62].

Overall, our data suggest that a molecular mapping could be a useful tool to investigate the brain and SC reorganization after SCI, considering the need to identify reliable and simple tools to evaluate efficacy of potential therapeutic approaches in preclinical models of SCI [21,22]. “Omics” technologies might significantly impact as screening approach for target identification. However, we consider that reliability of these bioinformatic approaches should be based on own data, or on data derived from databases sharing the full anatomical and functional characterization of the model.

## 4. Materials and Methods

### 4.1. Animals, Surgery and Care

CD-Sprague Dawley (Charles River, Calco, Italy) female rats of 200–250 gr (13 weeks old) were used for this study. All animal protocols described here were carried out according to the European Community Council Directives 86/609/EEC, approved by Italian Ministry of Health (D. Lgs 116/92, authorization n°574/2015-PR, 22 June 2015) as well as the European Community Council Directives 2010/63/UE. Moreover, animal protocols were carried out in compliance with the guidelines published in the ARRIVE and NIH Guide for the Care and Use of laboratory animals.

All animals were housed in pairs in plastic cages with standard bedding and diet ad libitum. One week before surgery all animals were handled and accustomed to bladder manipulation. Five animals for each experimental time were included in each experiment.

Animals underwent a contusive spinal lesion at the vertebral thoracic level (T9). Briefly, rats were pre-medicated with enrofloxacin and tramadol (4 mg/kg, s.c.), then anesthetized with isoflurane (1–3%) in O_2_. Rats were fixed in the stereotaxic table, and an incision with a length of 4 cm was made in the skin of the back. After cutting, the muscles were dissected layer by layer to fully expose the processus spinosus of T9–T11. The processus spinosus and lamina of T9 was removed by clamp to expose the spinal canal and spinal dura. Contusive lesion of the spinal cord was obtained with Impact One impactor (Leica BioSystems, Wetzlar, Germany) using a 2 mm tip with a force of 1 N (0.75 m/s) and 0 s of stance time; the depth of impact was 2 mm to reach ventral horns of gray matter. After performing SCI lidocaine (20 mg/mL) was administered topically. Back muscles were sutured, and the skin incision closed with wound clips. Upon completion of the surgery, animals received tramadol (4 mg/kg, s.c.) for 7 days as an analgesic and enrofloxacin (4 mg/kg, s.c.) for 7 days to prevent infection. Sham animals, receiving surgical treatment, laminectomy and pharmacological treatments without SCI, were used as control. Bladder were manually expressed twice a day until automatic voiding returned spontaneously. Animals were housed in single cage for the first week after surgery.

Evaluation of rat wellness was performed by body weight monitoring and a clinical score [63], evaluated daily for the first two weeks, then once a week until the day of sacrifice. In case of infection of lower urinary tract, animals were treated twice a day with enrofloxacin (4 mg/kg, s.c.) for three days. At the time of sacrifice, the tissues of interest (cerebral CTX-M, SCI tract T8−T11, SC segments caudal to the lesion T12−L3 and rostral to the lesion T7−T4) were dissected, and immediately snap frozen and stored at –80 °C till used.

### 4.2. BBB Score, Locomotion and Gait Analysis

For evaluation of hind limb functional locomotor loss, BBB score [16] was performed three DPL, and lesioned animals with a score greater than 1 were discarded from the analysis (4 animals in Lesion 8 days group and 1 animal in Lesion 45 days group were excluded from analysis). The BBB is a semiquantitative scale based on locomotor response of rats, that can take on values ranging from zero (no observable movement of the hindlimbs) to 21 (normal locomotion). BBB score was repeated once a week after surgery in both lesion, and sham animal groups to assess spontaneous motor recovery.

Gait analysis was performed with CatWalk XT (Noldus, Wageningen, The Netherlands) automatized system. Animals were trained before surgery to walk repeatedly along the platform, then tested two days before SCI and once a week after lesion. All animals underwent 3 compliant runs, as defined by instrument parameters (run duration from 0.5 to 8s), for each time point, and means of all parameters were calculated by CatWalk software. Gait analysis was performed on 11 different parameters, divided in 4 categories: Spatial Parameters (Print Area, Max Contact Area, Base of Support); Kinetic Parameters (Stand Time, Swing Time, Swing Speed, Single Stance); Comparative Parameters (Stride Length, Step Cycle); Coordination Parameters (Duty Cycle, Step Sequence Regularity Index). All parameters were analyzed for both hind paws, and front paws (represented as mean of right paw and left paw).

### 4.3. Histology

At the day of sacrifice rats were perfused with 4% paraformaldehyde and picric acid saturated aqueous solution in 0.1 M Sörensen buffer pH 7, then spinal cord tissue was dissected and post-fixed for 24 h, then washed with Sucrose 5% O/N. 14 μm thick sagittal and coronal sections were then prepared (Leica CM1950) and processed for histochemistry staining. Toluidine blue and Hematoxilin/Eosin (HE) were performed for evaluation of lesion area and inflammatory infiltrate. To define the area of lesion, sections were captured with Nikon Microphot–FXA equipped with a CCD camera Nikon DXM1200F (Nikon) and then measured with Photoshop (v. CS6; Adobe); all images were capture with a 4× magnification of the objective and reconstructed with Photoshop’s photomerge function. Lesion area was then determined for each reconstructed section with ImageJ software (NIH) as the number of pixels occupying the lesion site, and to obtain the ratio between lesion and healthy tissue, total section areas were measured for each section. 3D reconstruction was obtained aligning different levels of the same SC (sampling step 210 μm).

### 4.4. Total RNA Isolation, Reverse Transcription and PCR-Arrays

Right cerebral CTX-M, CB, BG, and SC segments (caudal and rostral to the lesion, and cervical) were homogenized, and total RNA isolation was performed using RNeasy Microarray Tissue Mini Kit (Qiagen, Hilden, Germany). Total RNA was eluted in RNase Free Water and using a spectrophotometer (Nanodrop 2000, Thermo Scientific, Waltham, MA, USA), absorbance values at 260, 280 and 320 were measured. RNAs were pooled for each group, to obtain a total amount of RNA of 0.5 µg and then retrotranscribed using the RT^2^ First Strand Kit (Qiagen). Synaptic Plasticity analysis was performed using Rat Synaptic Plasticity PCR Array (PARN-126ZA–Qiagen), a specific panel of 84 genes involved in Rat Synaptic Plasticity (Appendix A). Genes identified as housekeeping by the default asset of the array plate were included in the analysis due to observed gene expression regulation. Real time amplification of cDNA pools was achieved with CFX96 Real Time PCR System (Biorad, Hercules, CA, USA). Each array contains internal controls for genomic DNA contamination, for the quality of the amplification, and for the variability between plates. Thermal profile of PCR reactions was performed as follow: an activation step of Taq polymerase (95 °C, 10 min) followed by 40 cycles of denaturation (95 °C, 15 s) and annealing/extension (60 °C for 1 min and 30 s). At the end of the amplification cycles the dissociation curve was obtained by following a procedure consisting of first incubating samples at 95 °C for 1 min to denature the PCR-amplified products, then ramping temperature down to 65 °C and finally increasing temperature from 65 to 95 °C at the rate of 0.5 °C/s, continuously collecting fluorescence intensity over the temperature ramp. Analysis of genes expression of the array was performed using the RT^2^ Profiler PCR Array Data Analysis software v. 3.5 (SABiosciences, Qiagen, Frederick, MA, USA), and gene expression was normalized on reference genes suggested by the software (*Rplp1* for SC samples and *Hprt1* for brain samples).

### 4.5. Functional Pathway and Gene Set Enrichment Analysis

Time-series Gene Set Enrichment Analysis were performed by comparing the Pearson correlation between post-injury genes expression profiles. The complete gene lists for all comparisons are given in Appendix A. The resulting normalized gene sets expression values were used as input for GSEA desktop application version 4.1.0 (build 27) and tested for enrichment in selected molecular signatures database (MsigDB) collections: H collection: hallmark gene sets v7.4, C2 collection: CP: REACTOME database v. 7.4 and C5 collection: GO: BP database v. 7.4 [64,65]. The analysis was conducted with gene set parameters adjusted to a minimum size of 5 and a maximum size of 88 genes, to correct for the number of genes being tested. Any signature with absolute NES  ≥  1.5 and FDR adjusted *p*-value ≤ 0.1 were reported and used as input for GSEA Leading Edge Analysis (Appendix A). Mapping of pathway terms to pathway IDs was obtained via R v. 4.04 Bioconductor v3.12 GO.db v. 3.12.1 package and additional custom scripting [66,67]. Enrichment map visualizations (Figure 6 and Figure 7). were obtained through Cytoscape desktop application v. 3.8.2 and its ‘EnrichmentMap Pipeline Collection’ v. 1.1.0 application, with default parameters [68].

### 4.6. Statistical Analysis

Five animals per group were used for the experiments. Each data set was analyzed using two-way ANOVA, followed by post-hoc tests for exposure and genotype. Post tests were applied only for statistically significant two-way ANOVA interactions (*p* < 0.05). Student’s t-test was used for two-groups comparison. A probability level of *p* < 0.05 was considered to be statistically significant.

## Figures and Tables

**Figure 1 ijms-22-08606-f001:**
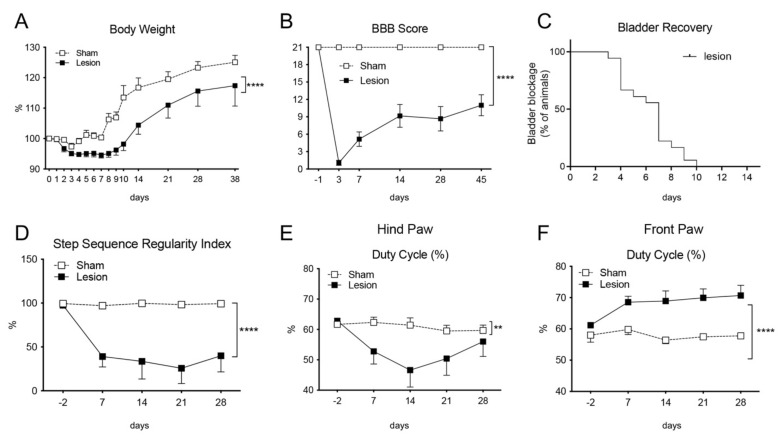
Animal monitoring of body weight (**A**), BBB scale of locomotion (**B**), spontaneous bladder function recovery, defined as percentage of animals showing bladder blockage during the observational time (**C**), step sequence regularity index (**D**) and duty cycle of the hind- (**E**) and front paws (**F**). Results are presented as mean ± SEM; statistical analysis: two-way ANOVA, ** *p* < 0.01; **** *p* < 0.0001.

**Figure 2 ijms-22-08606-f002:**
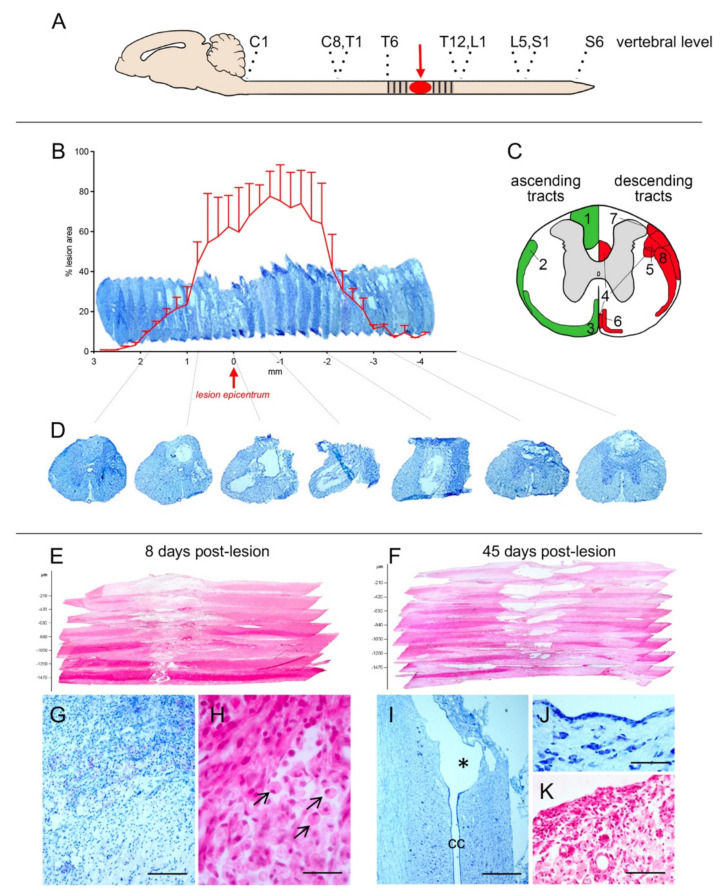
Anatomical analysis of the contusion model used in this study. (**A**) The epicenter of the impact (red) and the sampled area (dashed) are indicated. (**B**) 3D representation and quantitative evaluation of the lesion at 45 days post lesion (DPL), as performed in toluidine blue stained coronal sections. (**C**) Schematic representation of ascending (green) and descending (red) tracts in the Spinal Cord at T9. The numbers refer to the tracts, as follow: 1 dorsal column; 2 spinocerebellar; 3 spinothalamic; 4 corticospinal; 5 rubrospinal; 6 vestibulospinal; 7 reticulospinal; 8 raphespinal, (**D**) representative coronal sections all along the sampled rostro-caudal extension obtained from a sample animal. (**E**,**F**) 3D representation of the lesion d segments, as performed by HE staining on horizontal section at 8 (**E**) and 45 (**F**) DPL. (**G**–**K**) high power micrographs of the lesioned areas, illustrating the severe inflammatory cellular infiltrate (**G**) also enriched by gitter cells (marked by black arrows in (**H**)) at 8 DPL; the cavitation communication with the central canal (marked by * in (**I**)), respective ependymal-like lining layer (**J**), and persistence of the inflammatory cellular infiltrate (**K**) at 45 DPL. Bars: G 100 µm; I 25 µm; K 250 µm; J, K 200 µm.

**Figure 3 ijms-22-08606-f003:**
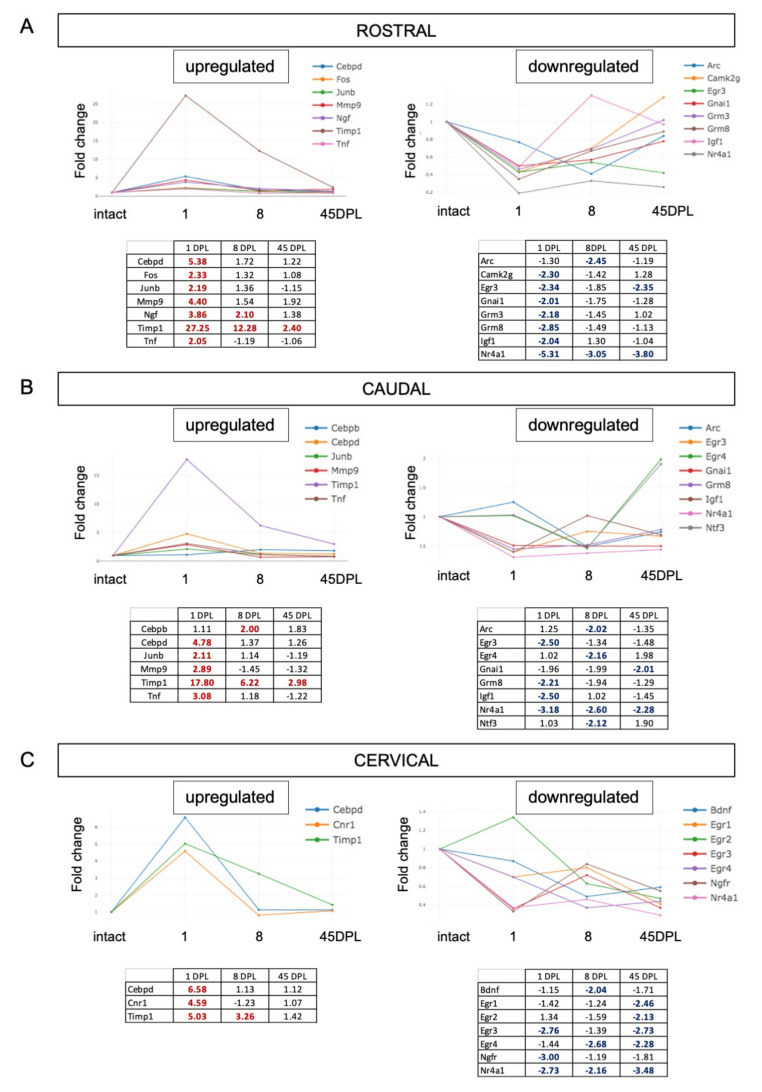
(**A**–**C**) Time course expression of upregulated and downregulated genes in Spinal Cord (SC) segments rostral (**A**) and caudal (**B**) to the lesion and in the cervical tract (**C**). Relative gene expression is shown as Fold Change (FC) normalized on SC segments from intact animals. Only genes with FC ≥ 2 are included in the figure.

**Figure 4 ijms-22-08606-f004:**
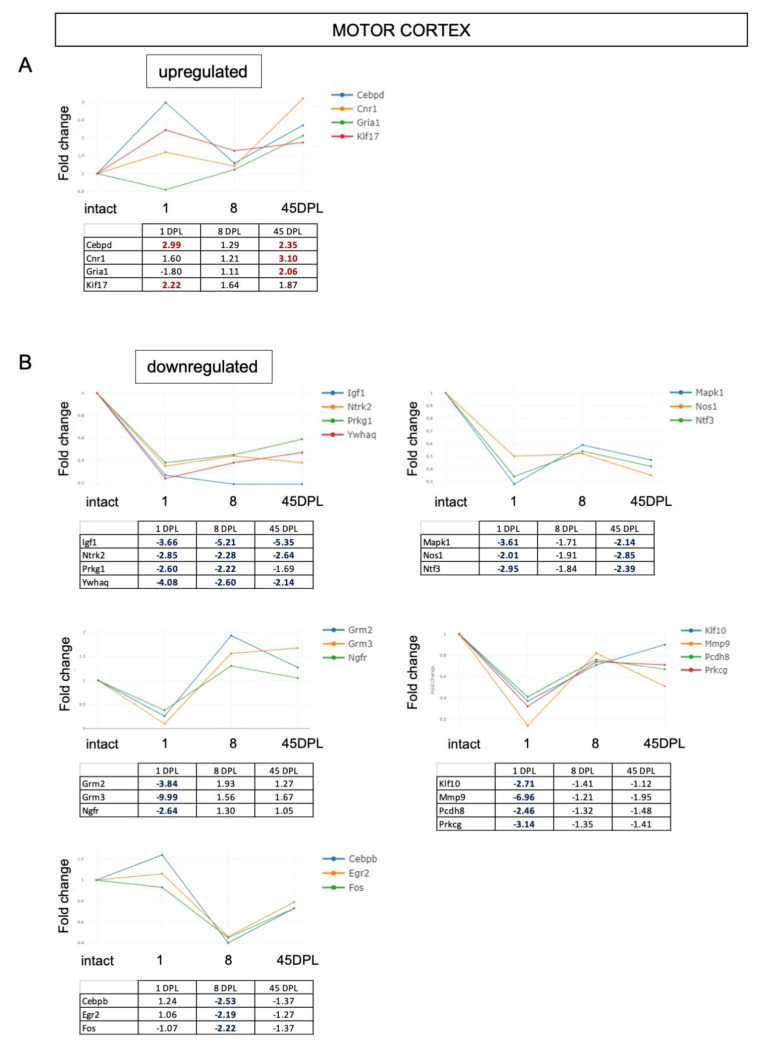
(**A**,**B**) Time course expression of upregulated (**A**) and downregulated (**B**) genes in motor cortex. Relative gene expression is shown as Fold Change (FC) normalized on motor cortex from intact animals. Only genes with FC ≥ 2 are included in the figure.

**Figure 5 ijms-22-08606-f005:**
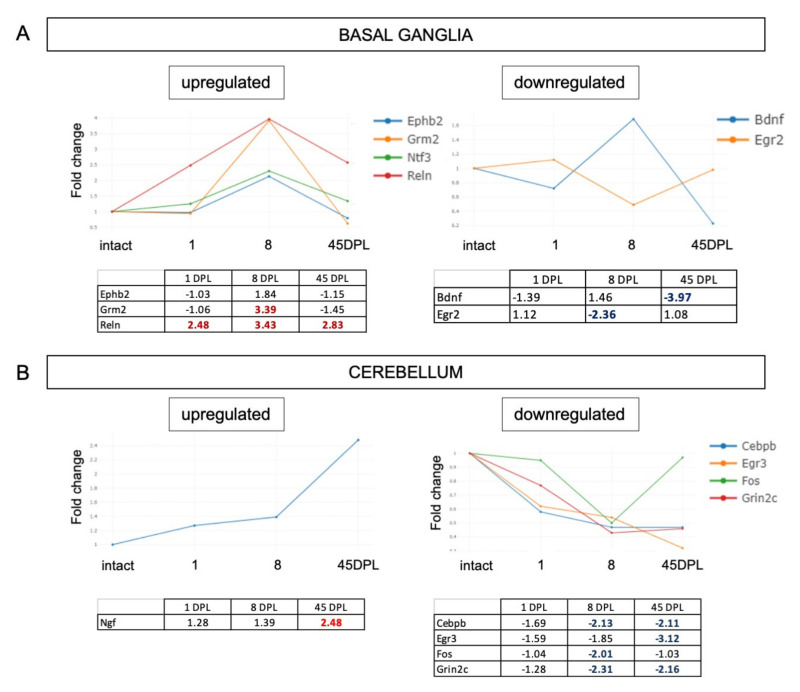
(**A**,**B**) Time course expression of upregulated and downregulated genes in Basal Ganglia (**A**) and Cerebellum (**B**). Relative gene expression is shown as Fold Change (FC) normalized on respective area from intact animals. Only genes with FC ≥ 2 are included in the figure.

**Figure 6 ijms-22-08606-f006:**
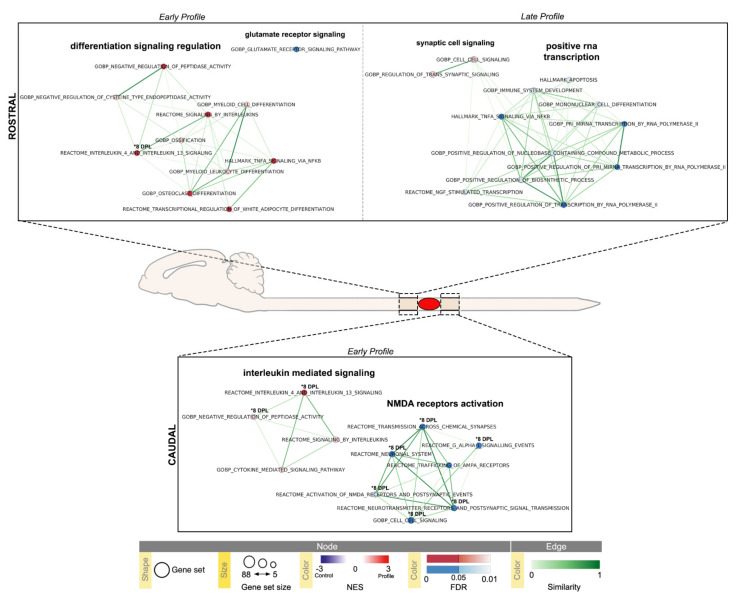
Time-series Gene Set Enrichment Analysis (GSEA) of Spinal Cord (SC) rostral and caudal segments visualized using Enrichment Map Cytoscape App. Results with absolute Normalized Enrichment Score (NES) ≥ 1.5 and False Discovery Rate (FDR) ≤ 0.1 are reported (SC rostral early 1–8 DPL and late 45 DPL profiles, Table 1; SC caudal early 1–8 DPL profiles, Table 2); pathways with FDR ≤ 0.05 are highlighted. Temporal profiles are labeled as Early Profile (peak at 1 DPL) and Late Profile (peak at 45 DPL); 8 DPL pathway enrichments (FDR ≤ 0.05) are individually marked (* 8 DPL). Pathways (circles) are connected by edges (lines) based on their similarity score and grouped into labeled clusters using AutoAnnotate application v.1.3.4. Red dots denote positively associated pathways (enriched respect to control, NES ≥ 1.5) and blue dots denote negatively associated pathways (decreased respect to control, NES ≤ −1.5). Edge width denotes overlap between pathways.

**Figure 7 ijms-22-08606-f007:**
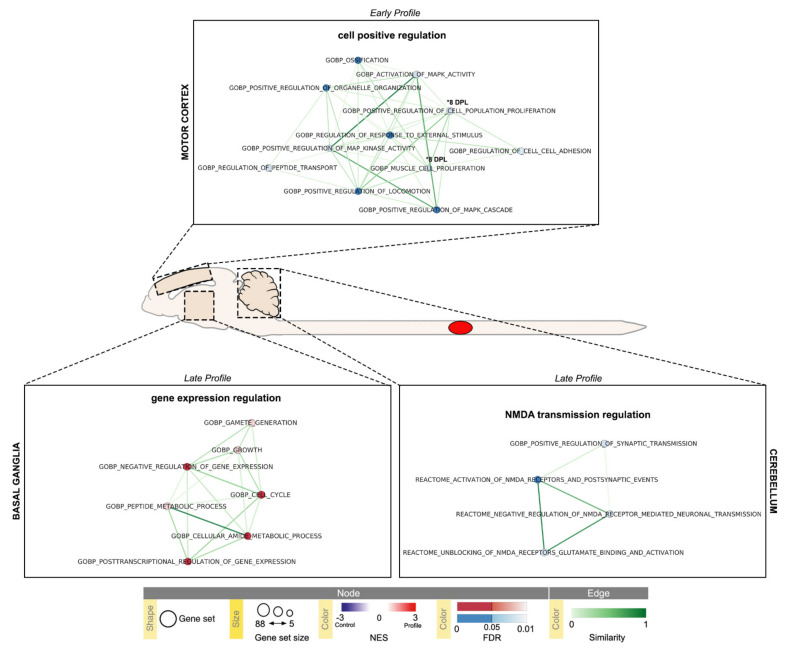
Time-series Gene Set Enrichment Analysis (GSEA) of cerebral Motor Cortex (CTX-M), Basal Ganglia (BG) and Cerebellum (CB) visualized using EnrichmentMap Cytoscape App. Results with absolute Normalized Enrichment Score (NES) ≥ 1.5 and False Discovery Rate (FDR) ≤ 0.1 are reported (CTX-M early 1–8 DPL profiles, and late 45 DPL profile, Table 3; BG late 45 DPL profile, Table 4; CB early 8 DPL, and late 45 DPL profiles Table 5); pathways with FDR ≤ 0.05 are highlighted. Temporal profiles are labeled as Early Profile (peak at 1 DPL) and Late Profile (peak at 45 DPL); 8 DPL pathway enrichments are individually marked. Pathways (circles) are connected by edges (lines) based on their similarity score and grouped into labeled clusters using AutoAnnotate application v.1.3.4. Red dots denote positively associated pathways (enriched respect to control, NES ≥ 1.5) and blue dots denote negatively associated pathways (decreased respect to control, NES ≤ −1.5). Edge width denotes overlap between pathways.

**Figure 8 ijms-22-08606-f008:**
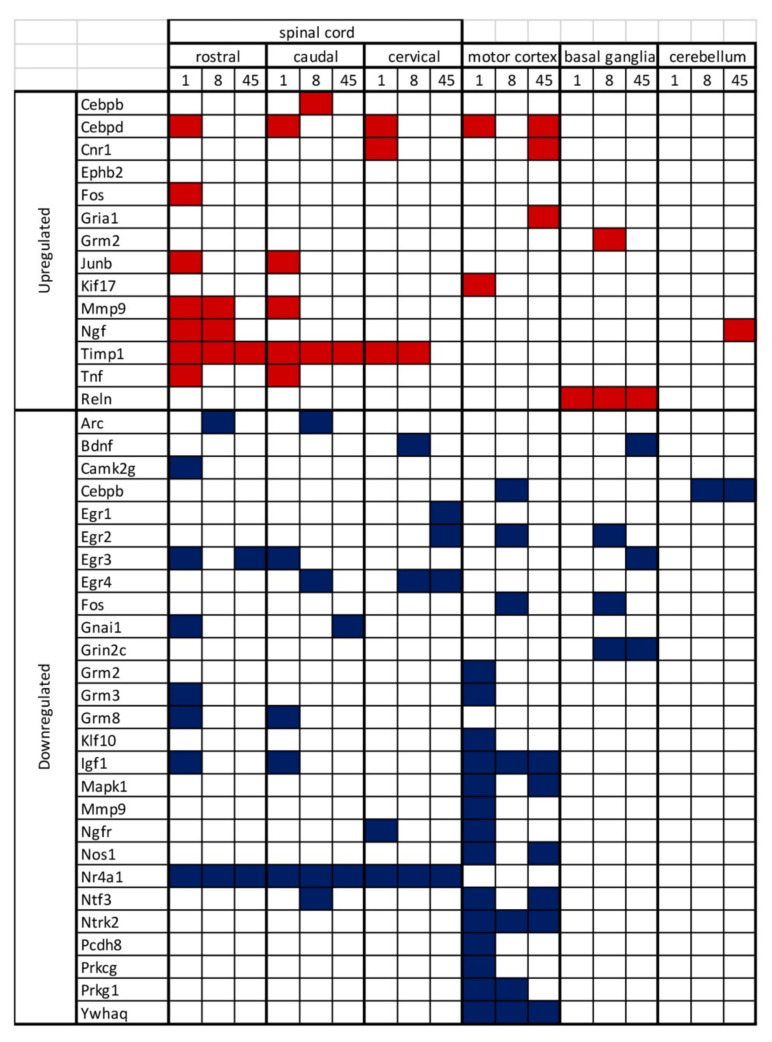
Summary of upregulated and downregulated genes for all the considered days post lesion and Central Nervous System areas.

**Table 1 ijms-22-08606-t001:** Early (1, 8 DPL peak) and late (45 DPL peak) profiles deregulated pathways in the Spinal Cord segment rostral to the lesion.

NAME	ID	SIZE	ES	NES	NOM.p.val	FDR.q.val	FWER.p.val
1 DPL							
REACTOME_INTERLEUKIN_4_AND_INTERLEUKIN_13_SIGNALING	R-RNO-6785807	8	0.922751	2.960325	0	0.001343 **	0.001
REACTOME_SIGNALING_BY_INTERLEUKINS	R-RNO-449147	15	0.59045	2.48295	0	0.02287 *	0.036
HALLMARK_TNFA_SIGNALING_VIA_NFKB	M5890	14	0.591473	2.428519	0	0.020713 *	0.048
GOBP_NEGATIVE_REGULATION_OF_PEPTIDASE_ACTIVITY	GO:0010466	6	0.83201	2.357697	0	0.026939 *	0.082
GOBP_OSTEOCLAST_DIFFERENTIATION	GO:0030316	6	0.807556	2.354578	0	0.021834 *	0.082
REACTOME_TRANSCRIPTIONAL_REGULATION_OF_WHITE_ADIPOCYTE_DIFFERENTIATION	R-RNO-381340	6	0.807932	2.300164	0	0.027823 *	0.125
GOBP_OSSIFICATION	GO:0001503	9	0.624718	2.155755	0.003165	0.069799	0.32
GOBP_NEGATIVE_REGULATION_OF_CYSTEINE_TYPE_ENDOPEPTIDASE_ACTIVITY	GO:2000117	5	0.793688	2.100962	0.00274	0.09092	0.439
GOBP_MYELOID_LEUKOCYTE_DIFFERENTIATION	GO:0002573	11	0.594451	2.088907	0.003676	0.086556	0.455
GOBP_MYELOID_CELL_DIFFERENTIATION	GO:0030099	13	0.502879	2.049668	0	0.097413	0.537
GOBP_GLUTAMATE_RECEPTOR_SIGNALING_PATHWAY	GO:0007215	17	−0.64627	−2.23902	0	0.031969 *	0.036
**8 DPL**							
REACTOME_INTERLEUKIN_4_AND_INTERLEUKIN_13_SIGNALING	R-RNO-6785807	8	0.784993	2.51836	0	0.021501 *	0.015
**45 DPL**							
GOBP_CELL_CELL_SIGNALING	GO:0007267	56	0.573257	2.007903	0	0.06429	0.117
GOBP_REGULATION_OF_TRANS_SYNAPTIC_SIGNALING	GO:0099177	40	0.549315	1.942583	0	0.081764	0.277
GOBP_POSITIVE_REGULATION_OF_NUCLEOBASE_CONTAINING_COMPOUND_METABOLIC_PROCESS	GO:0045935	28	−0.43293	−1.97778	0	0.082897	0.595
REACTOME_NGF_STIMULATED_TRANSCRIPTION	R-RNO-9031628	9	−0.66178	−2.01156	0.014235	0.069736	0.493
GOBP_IMMUNE_SYSTEM_DEVELOPMENT	GO:0002520	17	−0.50455	−2.01332	0	0.077424	0.489
HALLMARK_APOPTOSIS	M5902	6	−0.79285	−2.06413	0.003185	0.057833	0.346
GOBP_POSITIVE_REGULATION_OF_BIOSYNTHETIC_PROCESS	GO:0009891	27	−0.47244	−2.09493	0	0.05279	0.282
GOBP_MONONUCLEAR_CELL_DIFFERENTIATION	GO:1903131	7	−0.80153	−2.10801	0.003155	0.057605	0.261
GOBP_POSITIVE_REGULATION_OF_TRANSCRIPTION_BY_RNA_POLYMERASE_II	GO:0045944	24	−0.51624	−2.26319	0	0.013703 *	0.056
GOBP_POSITIVE_REGULATION_OF_PRI_MIRNA_TRANSCRIPTION_BY_RNA_POLYMERASE_II	GO:1902895	7	−0.80546	−2.26659	0	0.017004 *	0.052
GOBP_PRI_MIRNA_TRANSCRIPTION_BY_RNA_POLYMERASE_II	GO:0061614	7	−0.80546	−2.27329	0	0.022732 *	0.046
HALLMARK_TNFA_SIGNALING_VIA_NFKB	M5890	14	−0.76035	−2.75168	0	0 **	0

* FDR ≤ 0.05; ** FDR ≤ 0.005.

**Table 2 ijms-22-08606-t002:** Early (1, 8 DPL peak) profiles deregulated pathways in the Spinal Cord segment caudal to the lesion.

NAME	ID	SIZE	ES	NES	NOM.p.val	FDR.q.val	FWER.p.val
1 DPL							
REACTOME_INTERLEUKIN_4_AND_INTERLEUKIN_13_SIGNALING	R-RNO-6785807	8	0.9	2.823968	0	0 **	0
REACTOME_SIGNALING_BY_INTERLEUKINS	R-RNO-449147	15	0.571289	2.286192	0.003788	0.09028	0.118
GOBP_CYTOKINE_MEDIATED_SIGNALING_PATHWAY	GO:0019221	16	0.55529	2.273639	0	0.066719	0.129
GOBP_NEGATIVE_REGULATION_OF_PEPTIDASE_ACTIVITY	GO:0010466	6	0.816037	2.221286	0	0.0732	0.185
REACTOME_ACTIVATION_OF_NMDA_RECEPTORS_AND_POSTSYNAPTIC_EVENTS	R-RNO-442755	16	−0.61111	−1.96957	0	0.06652	0.439
REACTOME_TRAFFICKING_OF_AMPA_RECEPTORS	R-RNO-399719	11	−0.69351	−2.03043	0	0.03897 *	0.245
REACTOME_G_ALPHA_I_SIGNALLING_EVENTS	R-RNO-418594	18	−0.61807	−2.10975	0	0.01606 *	0.092
GOBP_CELL_CELL_SIGNALING	GO:0007267	56	−0.56237	−2.17293	0	0.00946 *	0.044
REACTOME_NEUROTRANSMITTER_RECEPTORS_AND_POSTSYNAPTIC_SIGNAL_TRANSMISSION	R-RNO-112314	22	−0.63534	−2.23874	0	0.006321 *	0.023
REACTOME_TRANSMISSION_ACROSS_CHEMICAL_SYNAPSES	R-RNO-112315	23	−0.65278	−2.33446	0	0.002479 **	0.006
REACTOME_NEURONAL_SYSTEM	R-RNO-112316	26	−0.6346	−2.35683	0	0.003253 **	0.004
**8 DPL**							
REACTOME_INTERLEUKIN_4_AND_INTERLEUKIN_13_SIGNALING	R-RNO-6785807	8	0.85	2.650963	0	0.004133 **	0.003
GOBP_NEGATIVE_REGULATION_OF_PEPTIDASE_ACTIVITY	GO:0010466	6	0.829268	2.291362	0	0.058247	0.082
REACTOME_SIGNALING_BY_GPCR	R-RNO-372790	22	−0.5614	−1.85086	0.003501	0.098777	0.748
REACTOME_ACTIVATION_OF_NMDA_RECEPTORS_AND_POSTSYNAPTIC_EVENTS	R-RNO-442755	16	−0.60603	−1.90244	0.002331	0.075281	0.55
GOBP_REGULATION_OF_SYNAPTIC_PLASTICITY	GO:0048167	21	−0.58122	−1.92569	0.001145	0.063086	0.455
REACTOME_G_ALPHA_I_SIGNALLING_EVENTS	R-RNO-418594	18	−0.61145	−1.95418	0	0.051105	0.361
GOBP_CELL_CELL_SIGNALING	GO:0007267	56	−0.54694	−1.97041	0	0.047453 *	0.304
GOBP_SENSORY_PERCEPTION	GO:0007600	9	−0.74845	−1.97867	0.001295	0.050751	0.278
GOBP_REGULATION_OF_TRANS_SYNAPTIC_SIGNALING	GO:0099177	40	−0.55088	−2.01597	0	0.036091 *	0.183
GOBP_RESPONSE_TO_ETHANOL	GO:0045471	5	−0.92069	−2.05396	0.001368	0.029496 *	0.128
REACTOME_NEUROTRANSMITTER_RECEPTORS_AND_POSTSYNAPTIC_SIGNAL_TRANSMISSION	R-RNO-112314	22	−0.6351	−2.10719	0	0.019659 *	0.066
REACTOME_TRANSMISSION_ACROSS_CHEMICAL_SYNAPSES	R-RNO-112315	23	−0.6471	−2.22689	0	0.004021 **	0.009
REACTOME_NEURONAL_SYSTEM	R-RNO-112316	26	−0.6586	−2.30701	0	0.003537 **	0.004

* FDR ≤ 0.05; ** FDR ≤ 0.005.

**Table 3 ijms-22-08606-t003:** Early (1, 8 DPL peak) profile of deregulated pathways in cerebral Motor Cortex.

NAME	ID	SIZE	ES	NES	NOM.p.val	FDR.q.val	FWER.p.val
1 DPL							
GOBP_POSITIVE_REGULATION_OF_CELL_POPULATION_PROLIFERATION	GO:0008284	21	−0.55747	−2.01551	0	0.074913	0.441
GOBP_ACTIVATION_OF_MAPK_ACTIVITY	GO:0000187	6	−0.8384	−2.01837	0	0.080351	0.431
GOBP_REGULATION_OF_CELL_CELL_ADHESION	GO:0022407	9	−0.75613	−2.06508	0.004202	0.055305	0.294
GOBP_POSITIVE_REGULATION_OF_LOCOMOTION	GO:0040017	11	−0.70714	−2.08796	0	0.049878 *	0.249
GOBP_POSITIVE_REGULATION_OF_MAP_KINASE_ACTIVITY	GO:0043406	6	−0.8384	−2.08928	0	0.056779	0.248
GOBP_OSSIFICATION	GO:0001503	9	−0.77164	−2.11155	0	0.0504 *	0.192
GOBP_POSITIVE_REGULATION_OF_MAPK_CASCADE	GO:0043410	9	−0.76861	−2.15364	0	0.039986 *	0.13
GOBP_POSITIVE_REGULATION_OF_ORGANELLE_ORGANIZATION	GO:0010638	10	−0.74741	−2.15712	0.002299	0.047046 *	0.122
GOBP_REGULATION_OF_PEPTIDE_TRANSPORT	GO:0090087	10	−0.73899	−2.15933	0	0.060828	0.118
GOBP_MUSCLE_CELL_PROLIFERATION	GO:0033002	8	−0.79286	−2.16342	0	0.089101	0.115
GOBP_REGULATION_OF_RESPONSE_TO_EXTERNAL_STIMULUS	GO:0032101	15	−0.68666	−2.28367	0	0.037286 *	0.025
**8 DPL**							
REACTOME_NGF_STIMULATED_TRANSCRIPTION	R-RNO-9031628	9	−0.70493	−1.96836	0.006316	0.096106	0.478
GOBP_MUSCLE_TISSUE_DEVELOPMENT	GO:0060537	11	−0.67464	−2.0182	0.002304	0.091098	0.334
GOBP_POSITIVE_REGULATION_OF_CELL_POPULATION_PROLIFERATION	GO:0008284	21	−0.57177	−2.03369	0	0.09632	0.289
GOBP_POSITIVE_REGULATION_OF_BIOSYNTHETIC_PROCESS	GO:0009891	27	−0.53552	−2.05973	0.002088	0.099952	0.235

* FDR ≤ 0.05.

**Table 4 ijms-22-08606-t004:** Late (45 DPL peak) profile deregulated pathways in the Basal Ganglia.

NAME	ID	SIZE	ES	NES	NOM.p.val	FDR.q.val	FWER.p.val
GOBP_CELL_CYCLE	GO:0007049	21	0.688508	2.187956	0	0.008712 *	0.007
GOBP_NEGATIVE_REGULATION_OF_GENE_EXPRESSION	GO:0010629	12	0.747671	2.080675	0	0.027282 *	0.043
GOBP_POSTTRANSCRIPTIONAL_REGULATION_OF_GENE_EXPRESSION	GO:0010608	8	0.837801	2.059416	0	0.022322 *	0.052
GOBP_CELLULAR_AMIDE_METABOLIC_PROCESS	GO:0043603	13	0.704689	2.000565	0	0.042499 *	0.13
GOBP_GROWTH	GO:0040007	14	0.678726	1.941892	0	0.075044	0.264
GOBP_GAMETE_GENERATION	GO:0007276	9	0.749268	1.908591	0	0.091444	0.364
GOBP_PEPTIDE_METABOLIC_PROCESS	GO:0006518	12	0.689573	1.903534	0.001361	0.082782	0.383

* FDR ≤ 0.05.

**Table 5 ijms-22-08606-t005:** Early (8 DPL peak) and late (45 DPL peak) profiles deregulated pathways in the Cerebellum.

NAME	ID	SIZE	ES	NES	NOM.p.val	FDR.q.val	FWER.p.val
8 DPL							
GOBP_RESPONSE_TO_TEMPERATURE_STIMULUS	GO:0009266	9	−0.83582	−1.94008	0	0.059114	0.056
**45 DPL**							
REACTOME_UNBLOCKING_OF_NMDA_RECEPTORS_GLUTAMATE_BINDING_AND_ACTIVATION	R-RNO-438066	12	−0.65789	−1.94966	0.004431	0.091273	0.444
REACTOME_NEGATIVE_REGULATION_OF_NMDA_RECEPTOR_MEDIATED_NEURONAL_TRANSMISSION	R-HSA-9617324	8	−0.7625	−1.95528	0.003063	0.097942	0.429
GOBP_POSITIVE_REGULATION_OF_SYNAPTIC_TRANSMISSION	GO:0050806	15	−0.66669	−2.06219	0	0.088657	0.138
REACTOME_ACTIVATION_OF_NMDA_RECEPTORS_AND_POSTSYNAPTIC_EVENTS	R-RNO-442755	16	−0.67115	−2.15044	0	0.04458 *	0.035

* FDR ≤ 0.05.

## Data Availability

Original data are available upon reasonable requests.

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
