# Peer review of "A Time-Course Study of the Expression Level of Synaptic Plasticity-Associated Genes in Un-Lesioned Spinal Cord and Brain Areas in a Rat Model of Spinal Cord Injury: A Bioinformatic Approach"

_ijms, 2021, doi:10.3390/ijms22168606_

Round 1
Reviewer 1 Report
The manuscript from Baldassarro & Sanna et al. provides a very interesting analysis of the molecular patterns and changes following SCI, both at different regions and at different time points. The manuscript is very well writen, with a clear message. This integrated analysis is novel and worth of publication. The introduction is clear and provides all the necessary context to understand the objectives. The hypothesis is clearly formulated and the design of the study answers the questions formulated. The methods are clearly explained, and ethical concerns regarding the use of animals are covered. Statistical analysis used is well described.
The discussion is very complete, but I would recommend to discuss the following topics:
1- The gene expression alterations observed are not cell specific. Given the increasing use of techniques such as single cell RNA seq, could you include in your discussion comparisons to recent gene expression studies post SCI or post traumatic injuries and speculate about which cells could be the main contributors for the results observed in your study?
2- Could you please elaborate more on the differences between rostral and caudal regions to the lesion? Are there any hints that could explain the results observed?
Moreover please review the following aspects:
1- Figure 1C – Please consider to invert the line in the graphic, so it is more comprehensible. Starting at 0% of bladder function/recovery and then finish at 100%
2- The Figure 1 legend is wrong, letters D and F should change with one another
3- Number 8 is missing in the legend of figure 2
4- Be careful with abbreviations throughout the manuscript. For instance, DPL does not seem to be explained anywhere in the manuscript. SCI or spinal cord injury is used interchangeably. The rule usually says that you introduce the abbreviation at first mention and then do not use anymore the extended version.
Other Details (by order of appearance):
Line 47 – Do you mean supraspinal inputs (instead of super-spinal)?
Line 50 – Lesion “side”, correct to “site”
Line 74 – “Highlight” in the singular form, instead of “highlights”
Line 96 – “Gait” pattern, instead of “gain” pattern
Line 104 – “Show” instead of “shown”
Line 134 – Figure 2 legend, “blue” instead of “bleu”
Line 160 – There is an extra “and” in the beginning of the line: “(…) downregulated at and 45 DPL.”
Line 160 – There is extra text, that is repeated from before. Check if the sentences are correct (it seems to be missing the word “caudal”)
Line 180 – You probably meant “downregulated” instead of “upregulated” (…and Ntf3 remain upregulated at 45 DPL)
Line 205 – “These” instead of “this”
Line 286 – Unnecessary repetition of the word “kinase”: “MAPK kinase”
Line 290 – Mention what pathway the code “GO:0033002” means
Line 393 – Correct: “The fine-tune of” instead of “the fine-tune the”
Line 414 – “Increase” instead of “increases”
Line 430 – “As well as” instead of “as well we” (repeated at line 594)
Line 439 – “Rats” instead of “rat”
If the authors proceed with these minor revisions, I would recommend publication.
Author Response
Reviewer 1
The manuscript from Baldassarro & Sanna et al. provides a very interesting analysis of the molecular patterns and changes following SCI, both at different regions and at different time points. The manuscript is very well writen, with a clear message. This integrated analysis is novel and worth of publication. The introduction is clear and provides all the necessary context to understand the objectives. The hypothesis is clearly formulated and the design of the study answers the questions formulated. The methods are clearly explained, and ethical concerns regarding the use of animals are covered. Statistical analysis used is well described.
We really thank the Reviewer for this general comment.
The discussion is very complete, but I would recommend to discuss the following topics:
1- The gene expression alterations observed are not cell specific. Given the increasing use of techniques such as single cell RNA seq, could you include in your discussion comparisons to recent gene expression studies post SCI or post traumatic injuries and speculate about which cells could be the main contributors for the results observed in your study?
We thank the Reviewer for rising this interesting point. Our study is, in fact, a first step in the analysis of the transcriptomic changes during the progression of the SCI using a data-driven approach with a specific pre-identified pathway of interest. We decided here to give more importance to the analysis of different sections of spinal cord and brain at different time points. Next steps will include high throughput quantification of the whole transcriptome (RNAseq) and single cell analysis to identify the molecular changes at cellular level.
When bulk RNAseq is used, a prevalence of the immunoresponse emerges from the pathway enrichment analysis of differentially expressed genes, both in early (1, 6 DPL) and late (28 DPL) phases. On the other hand, most of scRNA-seq are performed in mouse and huge differences have been described between mouse and rat lesion, especially in cavitation and scar formation, with the rat model described to be more similar to human SCI.
In the SCI progression, in the first day after lesion an innate immune response initiate by microglia is amplified my myeloid cell migrating to the injury site. After, around 3 DPL, OPCs are in the peak of the proliferative state and at 7 – 8 DPL numbers of macrophages and fibroblasts reach their peak, and astroglial scar starts to surround the fibrotic scar. Until 8 DPL local hypoxia stimulates angiogenesis and revascularization, with a peak around 14 DPL. ScRNA-seq in acute (1, 3 DPL) and sub-acute (7 DPL) phases in mouse model of SCI are able to define these steps in the SCI progression and to identify the cell population signature, according also to their activation state (Shi et al). In the acute and sub-acute phases the main response is mediated by resident and infiltrating immune system and OPC activation, this is in line with our observations of the extracellular matrix reorganization genes activation in the context of the synaptic plasticity pathway. Interestingly one of the main genes involved in the recovery identified in mouse SCI scRNAseq, Igf1, resulted downregulated in the acute phase in our rat model.
Now we added these comments and relative bibliography in the text (line 468).
2- Could you please elaborate more on the differences between rostral and caudal regions to the lesion? Are there any hints that could explain the results observed?
We thank the Reviewer for rising this interesting and still poorly understood point, which is important for both functional and maladaptive plasticity. A short discussion has been included (lines 410-422).
Moreover please review the following aspects:
1- Figure 1C – Please consider to invert the line in the graphic, so it is more comprehensible. Starting at 0% of bladder function/recovery and then finish at 100%
We apologize if the graph was not comprehensible, we now added a better explanation of the Y axis. The graph represents the bladder function recovery defined as the percentage of animals showing bladder blockage. Now this is stated in the figure legend and in the text (line 90).
2- The Figure 1 legend is wrong, letters D and F should change with one another
We apologize for the error in the figure legend, now fixed.
3- Number 8 is missing in the legend of figure 2
We apologize, this has been now included.
4- Be careful with abbreviations throughout the manuscript. For instance, DPL does not seem to be explained anywhere in the manuscript. SCI or spinal cord injury is used interchangeably. The rule usually says that you introduce the abbreviation at first mention and then do not use anymore the extended version.
We apologize for the mistakes; DPL is now written the first time we mentioned it (line 119) and the abbreviations are now homogenous throughout the text.
Other Details (by order of appearance):
Line 47 – Do you mean supraspinal inputs (instead of super-spinal)?
Line 50 – Lesion “side”, correct to “site”
Line 74 – “Highlight” in the singular form, instead of “highlights”
Line 96 – “Gait” pattern, instead of “gain” pattern
Line 104 – “Show” instead of “shown”
Line 134 – Figure 2 legend, “blue” instead of “bleu”
Line 160 – There is an extra “and” in the beginning of the line: “(…) downregulated at and 45 DPL.”
Line 160 – There is extra text, that is repeated from before. Check if the sentences are correct (it seems to be missing the word “caudal”)
Line 180 – You probably meant “downregulated” instead of “upregulated” (…and Ntf3 remain upregulated at 45 DPL)
Line 205 – “These” instead of “this”
Line 286 – Unnecessary repetition of the word “kinase”: “MAPK kinase”
Line 290 – Mention what pathway the code “GO:0033002” means
Line 393 – Correct: “The fine-tune of” instead of “the fine-tune the”
Line 414 – “Increase” instead of “increases”
Line 430 – “As well as” instead of “as well we” (repeated at line 594)
Line 439 – “Rats” instead of “rat”
We apologize for the typos and we thank the Reviewer for the fine revision of the text. All the errors have been fixed.
If the authors proceed with these minor revisions, I would recommend publication.
We thank the Reviewer for the comments and the suggestions.
Reviewer 2 Report
This is an interesting study presenting a molecular mapping of genes at different time points post-SCI in rats. This is done in multiple regions of the brain and spinal cord and brings interesting data about the reorganization of regions after injury.
Unfortunately, the font in most of the figures is very small and makes it impossible to read.
A few questions/comments:
Why did you only use females? Did you check for estrous cycle at the time of injury/euthanasia? What was the age of the female rats? This could potentially influence the results.
Could you define BBB score?
Line 533 you state that the number of animals is reported in the legends and in the results but I couldn't find it.
Author Response
This is an interesting study presenting a molecular mapping of genes at different time points post-SCI in rats. This is done in multiple regions of the brain and spinal cord and brings interesting data about the reorganization of regions after injury.
Unfortunately, the font in most of the figures is very small and makes it impossible to read.
We increased the size of the small font in the legends of graphs (Figure 3 – 5) and all the indications of the pathway enrichment analysis (Figure 6, 7).
A few questions/comments:
Why did you only use females? Did you check for estrous cycle at the time of injury/euthanasia? What was the age of the female rats? This could potentially influence the results.
In this study we use only CD adult female rats with 250-300grams of body weight at surgery (about 13 weeks old). Rodents are polyestrus mammal with a complete cycle every 4 or 5 days. We are aware that estrogens and estrous stages can modulate a wide range of neural functions, synaptic plasticity, cellular process, metabolism and trasport, signal trasduction and many other aspects (see for example Iqbal J., et al 2019). But is also known that estrous cycle disruption is a common phenomenon after Spinal Cord Injury (from 50 to 76% of cases), that remains for a long period (Shah PK et al., 2011; Shunmugavel A., et al 2012). For this reason we prefer not consider a specific estrous phase for control animals but we choose to pool the tissue samples to rebalance and integrate possible different estrous cycle stages between animals. We believe that this not compromized the validity of our results, but we consider correct to specify in the text this aspect to permit a better multi-factorial analysis in next studies (line 361-362).
We use only females. The choice to use both sexes is fundamental to study the impact of sex hormones and requires careful experimental design to consider all the other variables and keep them under control. But in this study we were not interested to evaluate sex differences. Despite that hormonal fluctuations during the estrous cycle could adds variability to research outcome with preclinical model of neurotrauma, female rodents remain the preferred sex to model SCI and the sex evaluated in approximately 70% of the pre-clinical literature (Stewart A., 2020). Male rodents show more severe post-operative complications and difficulty in manual emptying of the urinary bladder in case of temporary post-trauma neurological bladder. Another problem is the limit group housing of males due to size restrictions in the home cage, where single housing significantly decreases SCI recovery relative to group or environmentally enriched conditions (Berrocal Y., 2007).
- Shah PK, Song J, Kim S, Zhong H, Roy RR, Edgerton VR. Rodent estrous cycle response to incomplete spinal cord injury, surgical interventions, and locomotor training. Behav Neurosci. 2011 Dec;125(6):996-1002. doi: 10.1037/a0026032.
- Shunmugavel A, Khan M, Chou PC, Singh I. Spinal cord injury induced arrest in estrous cycle of rats is ameliorated by S-nitrosoglutathione: novel therapeutic agent to treat amenorrhea. J Sex Med. 2012 Jan;9(1):148-58.
- Stewart AN, MacLean SM, Stromberg AJ, et al. Considerations for Studying Sex as a Biological Variable in Spinal Cord Injury [published correction appears in Front Neurol. 2020 Oct 20;11:597689]. Front Neurol. 2020;11:802. Published 2020 Aug 5.
- Iqbal J, Tan ZN, Li MX, Chen HB, Ma B, Zhou X, Ma XM. Estradiol Alters Hippocampal Gene Expression during the Estrous Cycle. Endocr Res. 2020 Feb-May;45(2):84-101. doi: 10.1080/07435800.2019.1674868. Epub 2019 Oct 12. PMID: 31608702.
Could you define BBB score?
We thank the Reviewer for the indication of the missing definition. We now defined the BBB abbreviation the first time we mentioned the BBB score (line 88).
Line 533 you state that the number of animals is reported in the legends and in the results but I couldn't find it.
We apologize for the mistake, in fact the number of animals included in each experimental group was specified in the methods section, in the “4.1. Animals, surgery, and care” paragraph (line 492). Since the number of animals (5 per group) is the same for each experiment (both in single animal analysis, Figure 1 and 2, and gene expression analysis based on the pooled material, Figure 3 – 8) we now specified it in the “4.6 Statistical analysis” paragraph of the materials section and we removed the indicated sentence (line 641).
Round 2
Reviewer 2 Report
Changes have been made to improve the manuscript but some of the figures still contain fonts too small to be read.